# Protective Effects of Mitochondrial Uncoupling Protein 2 against Aristolochic Acid I-Induced Toxicity in HK-2 Cells

**DOI:** 10.3390/ijms23073674

**Published:** 2022-03-27

**Authors:** Chen Feng, Etienne Empweb Anger, Xiong Zhang, Shengdi Su, Chenlin Su, Shuxin Zhao, Feng Yu, Ji Li

**Affiliations:** Department of Clinical Pharmacy, School of Basic Medical Sciences and Clinical Pharmacy, China Pharmaceutical University, No. 639 Longmian Avenue, Jiangning District, Nanjing 211198, China; fengchen0504@163.com (C.F.); agepharm9@gmail.com (E.E.A.); zxzyzzzx@126.com (X.Z.); 3219091352@stu.cpu.edu.com (S.S.); suesunshine96@163.com (C.S.); zsx19950606@163.com (S.Z.); yufeng305cpu@163.com (F.Y.)

**Keywords:** aristolochic acid I (AA I), uncoupling protein 2 (UCP2), nephrotoxicity, apoptosis, oxidative stress

## Abstract

Aristolochic acid I (AA I) is one of the most abundant and toxic aristolochic acids that is reported to cause Aristolochic acid nephropathy (AAN). This paper was designed to assess whether mitochondrial Uncoupling Protein 2 (UCP2), which plays an antioxidative and antiapoptotic role, could protect human renal proximal tubular epithelial (HK-2) cells from toxicity induced by AA I. In this study, HK-2 cells were treated with different concentrations of AA I with or without UCP2 inhibitor (genipin). To upregulate the expression of UCP2 in HK-2 cells, UCP2-DNA transfection was performed. The cell viability was evaluated by colorimetric method using MTT. A series of related biological events such as Reactive Oxygen Species (ROS), Glutathione peroxidase (GSH-Px), and Malondialdehyde (MDA) were evaluated. The results showed that the cytotoxicity of AA I with genipin group was much higher than that of AA I alone. Genipin dramatically boosted oxidative stress and exacerbated AA I-induced apoptosis. Furthermore, the increased expression of UCP2 can reduce the toxicity of AA I on HK-2 cells and upregulation of UCP2 expression can reduce AA I-induced oxidative stress and apoptosis. In conclusion, UCP2 might be a potential target for alleviating AA I-induced nephrotoxicity.

## 1. Introduction

Plant-derived products or plants containing Aristolochic acids (AAs) such as Aristolochia debilis, A. contorta, A. manshuriensis and A. fangchi have been used for decades to treat a variety of diseases, including hepatitis, urinary tract infection, vaginitis, upper respiratory tract infection, eczema, and dysmenorrhea [1,2]. However, it has been reported that plants or botanical-derived products containing AAs can induce nephrotoxicity, and mutagenic and carcinogenic effects in humans [3,4,5,6,7]. The nephrotoxicity caused by AAs is known as aristolochic acid nephropathy (AAN). To date, the incidence of AAN is largely unknown and may probably be underestimated, as numerous ingredients known or suspected to contain AAs are used in traditional medicine in some countries, including China, Japan, India, Martinique, and Australia [8,9,10,11].

AAN is a kind of rapidly progressive interstitial nephritis characterized by increased serum creatinine, profound anemia, and mild tubular proteinuria, associated with histopathologic changes showing hypocellular interstitial infiltrate with severe fibrosis and tubular atrophy [12]. Despite the Food and Drug Administration and the International Agency for Research on Cancer (IARC) warning that the safety of botanical remedies containing AAs should be of concern [13,14], individuals are still exposed to AAs because of the worldwide distribution of Aristolochia plant species and the great use of herbal medicines in some regions [8,15]. However, there is no good treatment or medicine for AAN now.

Previous in vitro and in vivo studies have reported that AA-mediated nephrotoxicity is associated with oxidative stress, apoptosis, inflammatory process, and fibrosis [16]. However, the comprehensive cellular and molecular mechanisms by which AA-induced nephrotoxicity have not yet been completely elucidated and treatment of AAN is therefore limited. Mitochondrial uncoupling protein 2 (UCP2) is a member of the mitochondrial anion carrier proteins, which play a key role in regulation of membrane potential and cellular energy. UCP2 exerts antioxidant properties via uncoupling oxidative phosphorylation from ATP synthesis [17,18].

Several studies have reported antioxidative, anti-inflammatory, and antiapoptotic properties of UCP2. For example, Ding et al., using mouse model of acute kidney injury (AKI) of liposaccharide, demonstrated that over-expression of UCP2 in mouse alleviated LPS-induced ROS, inflammation and apoptosis [19]. Although there is the presence of other antioxidant systems in the regulation of oxidative stress, UCP2 intervention in decreasing ROS seems to be crucial because its effects take place in mitochondria. It has been suggested that UCP2 has a protective effect on ischemia/reperfusion (I/R)-induced AKI in mice by improving mitochondrial dynamics [20]. Similar results were reported by Zhong and Zhou [21,22]. Moreover, UCP2 expression increases in response to oxidative stress, which is implicated in several pathological conditions, such as AAN.

Aristolochic acid I (AA I) is highest in content in aristolochic acids, and has also been demonstrated to be the most representative component that can cause severe nephritis in vivo [23,24]. Nephrotoxicity is one of the most severe side effects of herbal medicines, which have gained much attention around the world [25,26]. To date, the mechanisms by which AA I induces nephrotoxic effects are not fully understood, so treatments for AAI-induced nephrotoxicity are also limited. Thus, this work was designed to assess whether UCP2, which plays an antioxidative and antiapoptotic role, could protect HK-2 cells from toxicity induced by AA I.

## 2. Results

### 2.1. Evaluation of Cell Viability, Morphological Changes and Cell Membrane Integrity Induced by AA I with or without Genipin on HK-2 Cells

AA I is a nephrotoxic and potent carcinogenic compound found in Aristolochiaceae plants (Figure 1A). Despite the measures prohibiting the use of products containing AAs, people are still exposed to them due to the widespread use of traditional products containing these compounds [8,15].

To assess its effects on cell viability, HK-2 cells were treated with different concentrations of AA I combined with or without genipin (UCP 2 inhibitor). The cell viability was assessed by MTT assay. As shown in Figure 1B, the combination of AA I and genipin increased the toxicity of AA I when compared to the treatment of AA I alone.

The morphological changes are one of the features of cytotoxicity. To investigate the toxicity of AA I on HK-2 cells, the cells were treated with 40 μM AA I with or without genipin (25 μM). As shown in Figure 2A, when AA I with genipin was used to treat cells, the cell morphology showed significant changes compared with the control group. The cell body showed different degrees of swelling or shrinkage, the number of adherent cells decreased or died, and the connections between cells were reduced.

LDH is a stable cytoplasmic enzyme in all cells and releases rapidly following plasma membrane damage and thus is used as an indicator of cytotoxicity [27]. To investigate the integrity of HK-2 cell membranes after treatment with AA I, LDH assay was performed. Table 1 shows that after treating HK-2 cells with AA I with or without genipin, the LDH content released by cells increased. Comparing each group, the content of LDH released by cells increased significantly in the AA I with genipin group. These data suggest that the toxicity of AA I to HK-2 may destroy the integrity of cell membrane and inhibition of UCP2 may increase the toxicity induced by AA I.

### 2.2. Evaluation of Apoptosis and Oxidative Status in HK-2 Cells by AA I with or without Genipin

Apoptosis was confirmed by analyzing the nuclear morphology of AA I-treated HK-2 cells. Nuclear morphology was evaluated with membrane-permeable blue Hoechst. To evaluate whether AA I-induced cytotoxicity was associated with apoptosis, we first examined such changes in the cell nuclei with Hoechst 33,258 staining. The cells showed typical morphological changes of apoptosis, such as nuclear shrinkage and chromatin condensation. The combination of AA I and genipin (25 μM) can exacerbate this situation (Figure 2B).

To evaluate the effects of AA I on intracellular ROS production, ROS was detected by DCFH-DA assay using fluorescence microscopy. Cells with increased ROS are green stained. As shown in Figure 2C, AA I increased ROS, while the combination with genipin drastically increased intracellular ROS. These results show that the toxicity of AA I to HK-2 cells may be due to oxidative stress and UCP2 inhibition would exacerbate oxidative stress, which could exacerbate the cytotoxicity of AA I.

The MDA level can be used as an important indicator of oxidative stress [28,29]. Glutathione peroxidase (GSH-Px) is an enzyme that plays an important role in protecting organisms from oxidative damage [29,30]. In order to investigate whether the cytotoxicity of AA I on HK-2 cells was related to oxidative stress, the level of MDA and GSH-Px was measured. As shown in Table 1, compared with control group or AA I group, MDA and GSH-Px level of AA I with genipin group was significantly increased, respectively. 

Caspase-3 plays an important role in the process of cell death and is usually used as a marker of apoptosis. As shown in Figure 3, the activities of Caspase-3 were significantly activated when treated with AA I or AA I with genipin.

### 2.3. Expression of UCP2 after AA I with or without Genipin on HK-2 Cells

In order to verify the role of UCP2 on AA I-induced cytotoxicity, the expression of UCP2 was determined after AA I with or without genipin was treated on HK-2 cells. Figure 4 showed that AA I had a certain inhibitory effect on the expression of UCP2, while the inhibitory effect was more intense when combined with UCP2 inhibitor. These results suggested that UCP2 may play an important role in AA I-induced cytotoxicity.

### 2.4. The Increased Expression of UCP2 Can Reduce the Toxicity of AA I on HK-2 Cells

To upregulate the expression of UCP2 in HK-2 cells, UCP2-DNA transfection using lipofectamine 2000 was performed. Figure 5 showed that UCP2 expression was successfully increased in HK-2 cells and it proved that UCP2-DNA transfection was successful.

As is shown in Figure 6A, compared with non-transfected cells, the HK-2 cells with high expression of UCP2 could significantly recover from AA I-induced cytotoxicity. Figure 7A,B show that the viability of transfected cells treated with different concentrations of AA I with or without genipin (25 μM) was significantly higher than that of non-transfected cells. In addition, the IC50 (Table 2) value of transfected cells treated with AA I with or without genipin was also higher than that of non-transfected cells. These results revealed that the toxicity of AA I on transfected cells was reduced.

### 2.5. The Increased Expression of UCP2 Can Reduce the Apoptosis and Oxidative Stress Induced by AA I on HK-2 Cells

The morphology of HK-2 cells was evaluated using Hoechst 33,258 staining. As shown in Figure 6, the apoptotic features of cells treated with AA I was improved on UCP2-DNA transfected cells. The intracellular ROS was measured using fluorescence microscopy with probe DCFH-DA. Cells with increased ROS are green-stained. The results showed that the generation of ROS in transfected cells was significantly decreased, which indicated that oxidative stress response was alleviated after increasing the expression of UCP2 in HK-2 cells.

The activity of caspase-3 (Figure 8A) was determined by caspase activity assays kits after treatment with AA I under different conditions for 24 h. The activity of capase-3 in UCP2-DNA transfection group was significantly decreased compared to the groups without UCP2-DNA transfection (*p* < 0.05). Meanwhile, the other related indicators, such as LDH, MDA, and GSH-Px activity, showed that there was a significant decrease in UCP2-DNA transfection group compared with the UCP2-DNA non-transfected group (Figure 8B–D). These results showed that under the condition of UCP2 high expression, the levels of apoptosis and oxidative stress induced by AA I were reduced, indicating that increasing UCP2 expression attenuated AA I-induced cytotoxicity on HK-2 cells.

## 3. Discussion

Herbal medicine involves the use of natural compounds, which have relatively complex active ingredients with varying degrees of side effects [31,32,33]. Many herbs which are found to cause nephrotoxicity exert toxic effects by inducing apoptosis in the renal tubular cells. In this paper, we demonstrated that the cytotoxicity of AA I was caused via apoptosis by Hoechst 33,258 staining and caspase-3. Cell apoptosis is accompanied by various morphological changes, including nuclear condensation, apoptotic bodies and DNA fragmentation [34,35], which was analyzed using Hoechst 33,258 staining (Figure 2B). Caspases are among the best characterized biochemical markers for apoptosis. The caspase activation is regulated through either extrinsic pathway (death receptor pathway) or intrinsic pathway (mitochondrial pathway) [36]. Both pathways converge on caspase-3 and subsequently on other proteases and nucleases that drive the terminal events of apoptosis [37]. Caspase-3 is the executioner caspase in the apoptosis mechanism [38]. Our study confirmed that caspase-3 was activated both in AA I group and combination group (AA I + genipin), indicating that AA I-induced apoptosis was caspase-dependent. (Figure 3).

Oxidative stress is an important factor for the induction of programmed cell death such as apoptosis, which regulates cell numbers and removes unwanted and potentially dangerous cells as a defense mechanism [34]. ROS are composed of hydroxyl radicals, superoxide anions, and their byproducts, which can cause lipid peroxidation, DNA fragmentation, and protein oxidation in multiple cell types [39]. MDA can aggravate the damage of cell membrane as the most important product of cell lipid oxidation that reflects the extent of damage of membrane system. GSH-Px is a low molecular scavenger, whose content is an important factor to evaluate organic antioxidative ability [40]. Our study demonstrated that AA I-treatment caused oxidative stress in HK-2 cells and increased intracellular ROS, MDA, and GSH-Px levels. Genipin was used as an inhibitor of UCP2, which aggravates the degree of oxidative stress induced by AA I (Table 1).

At present, the treatments of AA I-induced nephrotoxicity are very limited. Some studies have shown that inhibiting the uptake of AA I in the kidney or enhancing the metabolism of AA I in the liver and kidney can reduce the nephrotoxicity of AA I. Baudoux et al. observed that in vitro use of probenecid, a drug for treating gout, restricted the entry of AA I into renal tubules by inhibiting organic anion transporters [41]. The main human enzymes activating AA-I are NAD (*p*) H:quinone oxidoreductase (NQO1), cytochrome P450 (CYP) 1A1/2, NADPH:CYP reductase (POR) and protaglandin H synthase (COX) [42]. Tanshinone I, an inducer of CYP1A1, has been tested on a C57BL/6 mouse model of AAN. Pretreatment with tanshinone I has shown protective effects against AA-induced acute kidney injury, notably through an increase in AA metabolization by CYP1A1 [43]. Dicoumarin is called an NQO1 inhibitor. Dicoumarol suppressed the nitroreduction of AA I and the formation of aristolactam I, which in turn can be converted into toxic metabolite in renal tissue [44]. Current studies are focused on the inhibition of transporters and metabolic enzymes, but the use of enzyme inducers or competitive inhibition of transporters would affect the absorption and metabolism of other drugs, resulting in other adverse reactions. Therefore, treatments of nephrotoxicity caused by AA I need further study.

UCP2 is the most well-known protein in the UCP family, which plays an important role in decreasing ROS production and regulating mitochondrial apoptosis signals [21,22,45]. In this study, we found that both AA I and genipin significantly inhibited the expression of UCP2 (Figure 4) and the inhibition of UCP2 increased oxidative stress and exacerbated AA I-induced apoptosis. However, after the expression of UCP2 was upregulated, the degree of AA I on ROS production and apoptosis decreased. These results confirmed the role of UCP2-inhibited ROS production and reduced cell apoptosis (Figure 6).

In conclusion, the study revealed that the toxicity of AA I on HK-2 cells was caused by apoptosis and oxidative stress. Inhibiting the expression of UCP2 aggravated the toxicity, while increasing the expression of UCP2 reduced the cytotoxicity caused by AA I. For the first time, the study documented that AA I-induced apoptosis and oxidative stress were related to UCP2. UCP2 upregulation could reduce AA I-induced apoptosis and alleviate oxidative stress. Thus, UCP2 might be a potential target for alleviating nephrotoxicity induced by AA I.

## 4. Materials and Methods

### 4.1. Cell Culture and Materials

Human renal proximal tubular epithelial cells (HK-2 cells) were kindly provided by Shanghai Institute of Materia Medica (China) and cultured in DMEM/F12 culture medium containing 10% fetal bovine serum (FBS) and 1.0% penicillin–streptomycin solution (Gemini Bio Products, West Sacramento, CA, USA).

Aristolochic acid I (AA I) (Purity HPLC ≥ 98%), Dimethyl sulfoxide (DMSO), Hoechst 33,258 were purchased from Solarbio (Solarbio Science & Technology Co., Ltd., Beijing, China); genipin (purity > 98%) was purchased from Sigma. The 3-(4,5-dimethyl thiazol-2-yl-)-2,5-diphenyl tetrazolium bromide (MTT), lactate dehydrogenase (LDH) assay kit, lipid peroxidation (MDA) kit, and glutathione peroxidase (GSH-Px) were sourced from Nanjing Jiancheng Bioengineering Institute, Nanjing, China. Caspase-3 assay kit was purchased from Beyotime Biotechnology, Beijing, China. UCP2 polyclonal antibody (Cat # PA5-80203, Lot # AA2D12N), ECL solution and prestained protein marker were obtained from Thermo Fisher Scientific, Shanghai, China. To upregulate the expression of UCP2 in HK-2 cells, UCP2 DNA transfection using lipofectamine 2000 (Thermo Fisher Scientific, Shanghai, China) was performed.

### 4.2. Cell Viability Assay

The cell viability was assessed using the MTT assay. HK-2 cells were seeded into 96-well plate at a density of 1.5 × 10^4^ cells/well. After 24 h of incubation, cells were treated with different concentrations of AA I with or without genipin (25 μM) for 24 h. Then 5 mg/mL of MTT in PBS was added into each well with the final concentration at 0.5 mg/mL. After 4 h of incubation at 37 ℃ in the dark, the culture supernatants were removed from the wells and DMSO was added into each well to dissolve the purple formazan crystals. The absorbance at 490 nm was measured by using microplate reader (Thermo Fisher, Multiskan GO, Waltham, MA, USA).

### 4.3. Examination of Morphological Changes

Cells undergoing apoptosis display typical common characteristics such as cell shrinkage, nuclear condensation, membrane blebbing, chromatin cleavage, and formation of pyknotic bodies of condensed chromatin. The determination of the morphological changes was visualized using inverted phase contrast microscope (DFC45C, Leica microsystems Ltd., Wetzlar, Germany). Cells were treated with different concentrations of AA I with or without genipin and incubated for 24 h, then observed under inverted contrast microscope.

### 4.4. Hoechst 33,258 Staining for Apoptotic Cells

Hoechst 33,258 is a fluorescent dye that binds to the regions of DNA rich in adenine–thymine. It forms with DNA a complex, which is brightly fluorescent, and can be easily detected by fluorescent microscope. HK-2 cells were seeded on glass coverslips in 6-well plates and treated with various concentrations of AA I with or without genipin for 24 h. After treatment, the cell medium was discarded and cells were washed with PBS and fixed with 4% paraformaldehyde for 15 min at 37 °C. Then the fixed cells were washed with PBS and stained with Hoechst 33,258. The cells were then incubated in the dark for 5 min at 37 °C, and photographed with a fluorescence microscope (Olympus IX70 Inverted Fluorescence Phase Contrast Microscope, Tokyo, Japan).

### 4.5. Detection of Caspase-3 Activity Assay

The activity of caspase-3 was measured using caspase apoptosis detection kit according to the manufacturer’s instructions. HK-2 cells at a density of 1.5 × 10^6^ were seeded in 6-well plate for 24 h. The cells were treated with different concentrations of AA I with or without genipin for 24 h. Then the cells were washed with cold PBS and resuspended in 100 µL of chilled cell lysis buffer, then incubated on ice for 10 min. The lysates were centrifuged at 4 °C at 12,000× *g* for 10 min. After determination of protein concentrations using BCA assay kit, 50 μL of supernatants (~30 μg) were added in working solutions. After incubations for 60 min at 37 °C, the absorbance was measured at 405 nm using a microplate reader.

### 4.6. Reactive Oxygen Species (ROS) Assay

The intracellular ROS was measured by fluorescence-based assay. This assay consists of the ability of the cell-permeant reagent 2, 7dichlorofluorescin diacetate (DCFH-DA) dye to be converted by cellular esterases to a non-fluorescent compound, which is later oxidized by ROS into 2, 7dichlorofluorescein (DCF). Fluorescent DCF can be detected by fluorescence spectroscopy with excitation/emission at 485 nm/535 nm. HK-2 cells were seeded on glass coverslips in 6-well plates for 24 h, and then treated with AA I or AA I in combination with genipin for 24 h. Following the treatment, cells were washed two times with PBS and incubated with DCFH-DA at the final concentration of 10 μM in cell-free serum medium in the dark for 30 min at 37 ℃. The cells were washed with PBS. The ROS was measured by fluorescence microscope.

### 4.7. Lactate Dehydrogenase (LDH) Assay

Cell death was evaluated by the quantification of plasma membrane damage which resulted in the release of lactate dehydrogenase (LDH). HK-2 cells were treated with different concentrations of AA I with or without genipin for 24 h. After the treatment, the cells were washed with PBS and treated with cell lysis solution. After centrifugation, the supernatants were collected. The level of LDH released in the cell culture supernatant was detected by LDH assay (Jiancheng Bioengineering Institute, Nanjing, China) detection kit following the manufacturer’s instructions.

### 4.8. Lipid Peroxidation (MDA) Assay

Malondialdehyde (MDA) and 4-hydroxynonenal (4-HNE) are natural bi-products of lipid peroxidation. MDA reacts with thiobarbituric acid (TBA) to generate an MDA-TBA complex. This complex can be easily quantified calorimetrically (OD = 532 nm). HK-2 cells were seeded in 6-well plates at density of 1.5 × 10^6^ cells/wells foe 24 h. After treatment for 24 h, cells were harvested and cell lysates were centrifuged and supernatant was collected for assay following the manufacturer’s instructions.

### 4.9. Glutathione Peroxidase (GSH-Px) Assay

GSH-Px reduces cumene hydroperoxide, and oxidizes GSH to GSSG. The generated GSSG is reduced to GSH with consumption of NADPH by glutathione reductase. The decrease of NADPH is proportional to GSH-Px activity in the reactions. The decrease of NADPH can be easily measured at 340 nm. HK-2 cells were seeded in 6-well plates at density of 1.5 × 10^6^ cells/wells for 24 h. After treatment for 24 h, cells were harvested, cell lysates were centrifuged, and supernatant was collected for assay following the manufacturer’s instructions.

### 4.10. Cell Transfection

To upregulate the expression of UCP2 in HK-2 cells, UCP2-DNA transfection using lipofectamine 2000 was performed. Before treatment with AA I or AA I + genipin, HK-2 cells were incubated for 24 h with the complex UCP2 DNA-Lipofectamine 2000 in reduced serum medium (Opti-MEM I Reduced Serum Medium, Thermo Fisher Scientific). To be specific, 4 μg of UCP2-DNA (Sino bilogical, Beijing, China) were diluted into 200 μL of Opti-MEM I Reduced Serum Medium (Thermo Fisher Scientific, Shanghai, China), and then 10 μL of lipofectamine 2000 were diluted in 200 μL of Opti-MEM I Medium, and then incubated for 5 min at room temperature. After 5 min of incubation, the diluted UCP2-DNA was combined with the diluted Lipofectamine 2000. After 20 min of incubation at room temperature, the 400 μL of DNA-Lipofectamine 2000 complexes were added to each well, and then the cells were incubated at 37 ℃ in a CO_2_ incubator for 24 h for transgene expression.

### 4.11. Western Blotting Analysis

The extracted protein samples were mixed with 2× SDS loading buffer (100 mM Tris-Cl, pH 6.8; 4% *w/v* of sodium dodecyl sulfate (SDS); 0.2% *w/v* bromophenol blue; 20% *v/v* of glycerol, 200 mM dithiothreitol), and boiled for 5 min. Proteins were separated by SDS-polyacrylamide gel electrophoresis and transferred onto a nitrocellulose membrane. Membranes were blocked with 5% free-fat milk in Tris-Buffered Saline, TWEEN^®^ 20 (TBST) buffer for 2 h, and then incubated with specific primary antibodies overnight at 4 ℃. After washing three times with TBST buffer, the membranes were incubated with secondary antibodies for 2 h, and then washed three times with TBST buffer. Immune-reactive proteins were detected using an ECL detection kit, and visualized.

### 4.12. Statistical Analysis

All data were presented as mean ± SD from at least 3 independent experiments and analyzed with SPSS 19.0 software (SPSS, Chicago, IL, USA). Differences were evaluated by Student’s *t*-test analysis. Statistical significance was defined as *p* < 0.05.

## 5. Conclusions

Our study revealed that the toxicity of AA I to HK-2 cells was dose-dependent and the integrity of cell membrane was damaged after HK-2 cells were treated with AA I. Honchest staining revealed that AA I resulted in HK-2 cell nuclear contraction and chromatin condensation, suggesting that cells were in a state of apoptosis. Caspase-3 also confirmed such results. ROS generation and other oxidative stress indicators indicated that cells were under oxidative stress. We also demonstrated that genipin, a UCP2 inhibitor, aggravated AA I-induced toxicity, apoptosis, and oxidative stress. Importantly, after we increased the expression of UCP2, the cytotoxicity of AA I + genipin, as well as the degree of apoptosis and oxidative stress, was significantly reduced. Therefore, lowering UCP2 expression can aggravate AAI-induced cytotoxicity, but increasing UCP2 expression can alleviate AA I-induced cytotoxicity. UCP2 has a protective effect against AA I-induced cytotoxicity.

## Figures and Tables

**Figure 1 ijms-23-03674-f001:**
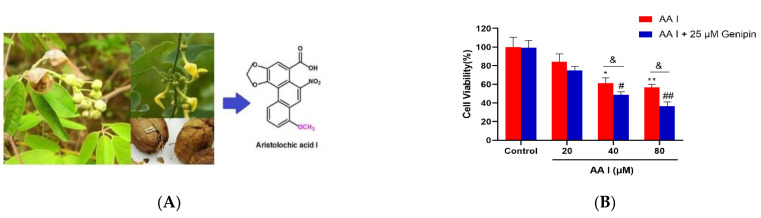
(**A**) Aristolochiaceae family and the chemical structure of main Aristolochic acids components AA I. (**B**) Toxicity of AA I to HK-2 cells. MTT was used for cell viability and cytotoxicity. HK-2 cells were treated with different concentrations of AA I with or without genipin for 24 h. All the experiments were repeated at least three times, and the data were expressed as means ± SD (* *p* < 0.05; ** *p* < 0.01 vs. AA I Control group; ^#^
*p* < 0.05; ^##^
*p* < 0.01 vs. AA I + 25 μM Genipin Control group; ^&^
*p* < 0.05).

**Figure 2 ijms-23-03674-f002:**
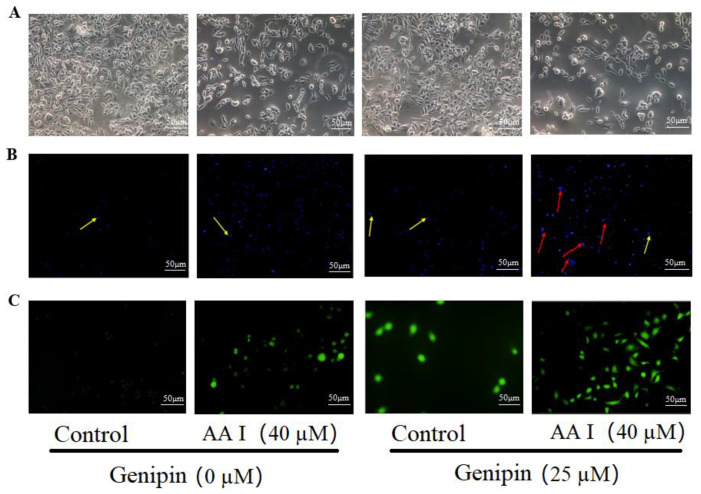
The toxicity of AA I to HK-2 cells was due to cell apoptosis and oxidative stress. (**A**) Inverted phase contrast microscope was used to observe cellular morphology (scale bar, 50 µM). (**B**) The nuclear morphology of HK-2 cells was evaluated using Hoechst 33,258 staining by a fluorescence microscope (scale bar, 50 µM). The yellow arrows indicate the chromatin condensation, which is brightly stained and the red arrows show DNA fragmentation. (**C**) The intracellular ROS was measured using fluorescence microscopy (scale bar, 50 µM) with probe DCFH-DA. HK-2 cells were treated with 40 μM AA I or 25 μM genipin for 24 h.

**Figure 3 ijms-23-03674-f003:**
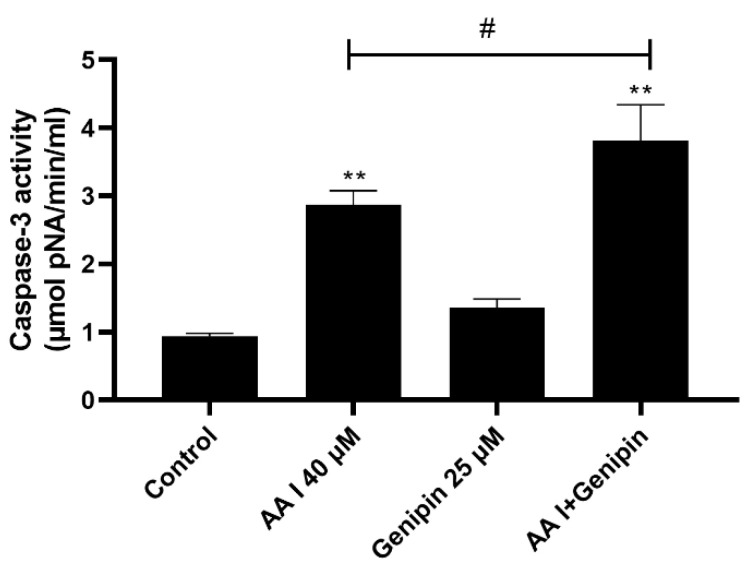
The activity of caspase-3 was determined by caspase activity assays kits after treatment with 40 μM AA I or 25 μM genipin for 24 h. All the experiments were repeated at least three times, and the data were expressed as means ± SD (** *p* < 0.01 vs. Control group; ^#^
*p* < 0.05).

**Figure 4 ijms-23-03674-f004:**
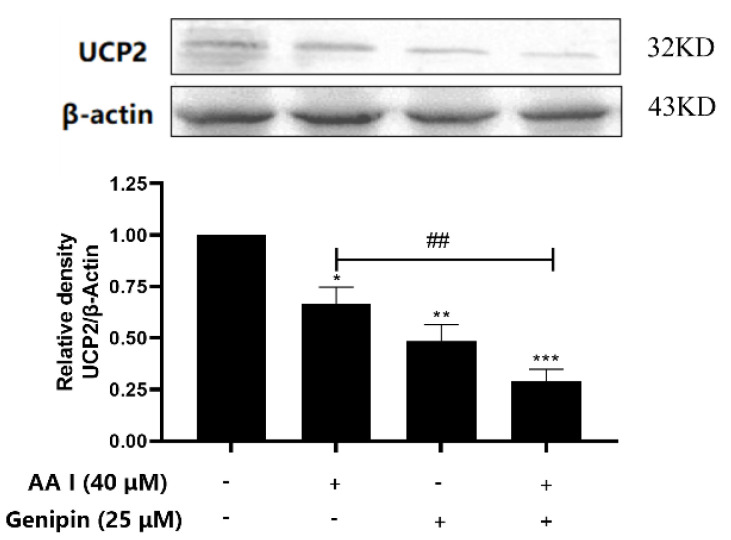
UCP2 expression in HK-2 cells with different treatments was detected by Western blotting. Each bar represents the mean ± SD of three independent experiments (* *p* < 0.05; ** *p* < 0.01; *** *p* < 0.001 vs. Control group; ^##^
*p* < 0.01).

**Figure 5 ijms-23-03674-f005:**
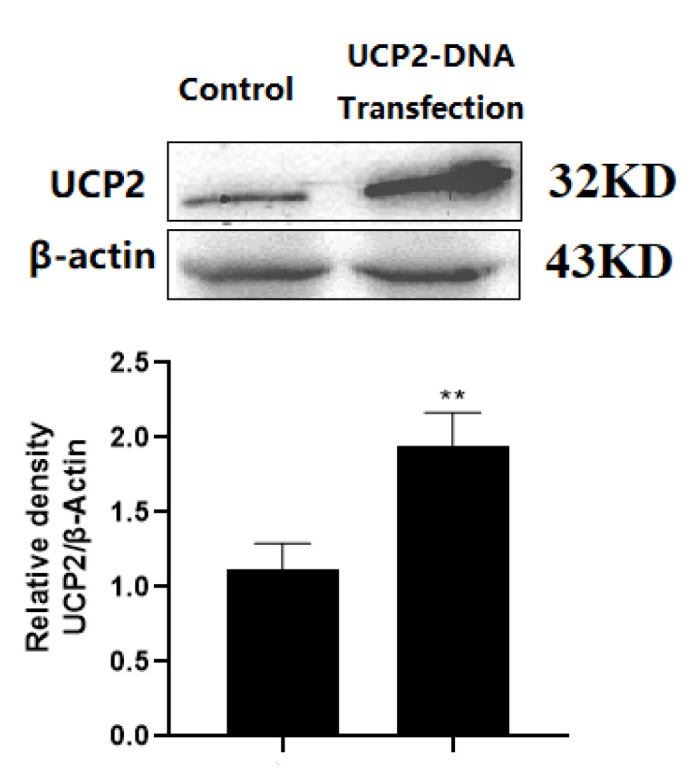
UCP2 expression was shown in both transfected and non-transfected cells. Each bar represents the mean ± SD of three independent experiments (** *p* < 0.01 vs. Control group).

**Figure 6 ijms-23-03674-f006:**
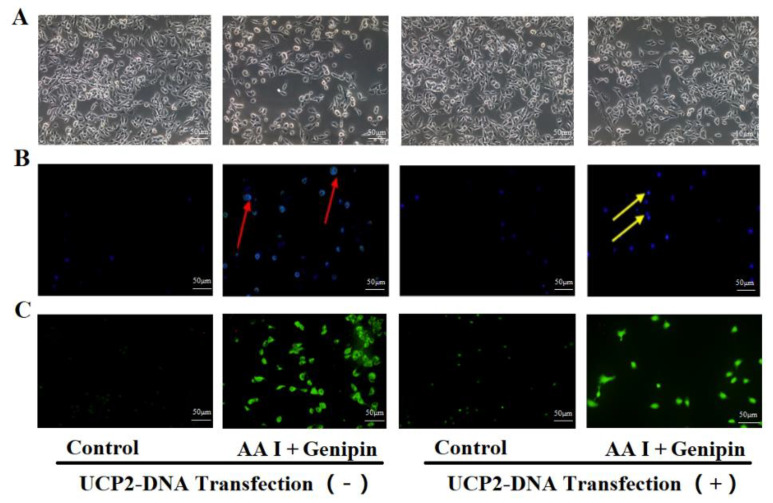
Increasing UCP2 expression alleviated the cytotoxicity of AA I to HK-2 cells. (**A**) Inverted phase contrast microscope was used to observed the morphology of transfected or non-transfected cells after cells were treated with AA I and genipin (scale bar, 50 µM). (**B**) The nuclear morphology of transfected or non-transfected cells was evaluated using Hoechst 33,258 staining by fluorescence microscope (scale bar, 50 µM). The yellow arrows indicate the chromatin condensation, which is brightly stained and the red arrows show DNA fragmentation. (**C**) The intracellular ROS was measured using fluorescence microscopy (scale bar, 50 µM) with probe DCFH-DA.

**Figure 7 ijms-23-03674-f007:**
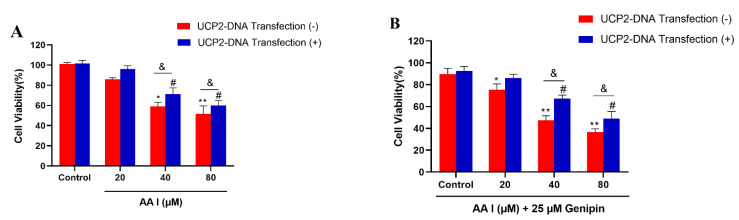
MTT was used for cell viability and cytotoxicity. (**A**) HK-2 cells with or without UCP 2-DNA transfection were treated with different concentration of AA I for 24 h. (**B**) HK-2 cells with or without UCP2-DNA transfection were treated with different concentration of AA I + genipin for 24 h. Each bar represents the mean ± SD of three independent experiments (* *p* < 0.05; ** *p* < 0.01; vs. UCP2-DNA Transfection (−) Control group; ^#^
*p* < 0.05 vs. UCP2-DNA Transfection (+) Control group; ^&^
*p* < 0.05).

**Figure 8 ijms-23-03674-f008:**
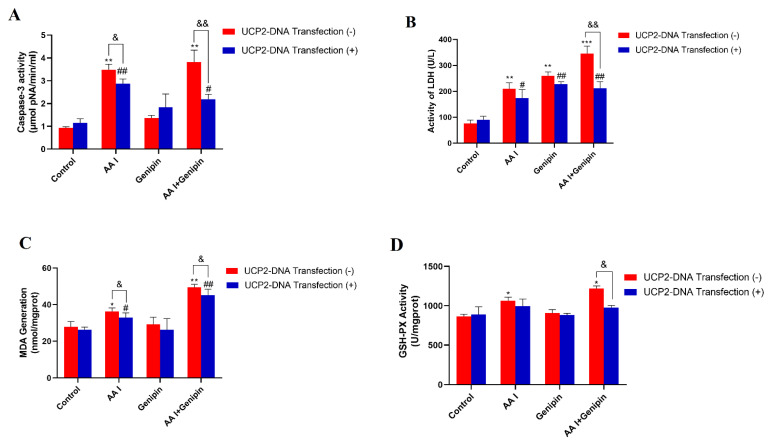
Increasing UCP2 expression alleviated AAI-induced apoptosis and oxidative stress. HK-2 cells with or without UCP2-DNA transfection were treated with 40 μM AA I + 25 μM genipin for 24 h. Caspase-3 activity (**A**), LDH activity (**B**), MDA generation (**C**) and GSH-Px activity (**D**) were detected on transfected or non-transfected cells. Each bar represents the mean ± SD of three independent experiments (* *p* < 0.05; ** *p* < 0.01; *** *p* < 0.001 vs. UCP2-DNA Transfection (−) Control group; ^#^
*p* < 0.05; ^##^
*p* < 0.01 vs. UCP2-DNA Transfection (+) Control group; ^&^
*p* < 0.05; ^&&^ *p* < 0.01).

**Table 1 ijms-23-03674-t001:** Values of LDH activity, MDA activity and GSH-Px activity.

Parameter	Control	AA I	Genipin	AA I + Genipin
LDH (U/L)	75.56 ± 13.88	260.14 ± 14.74 *	210.00 ± 22.47 *	345.57 ± 28.83 *
MDA(nmol/mgprot)	27.80 ± 3.00	36.21 ± 1.94 *	29.29 ± 3.81	49.37 ± 0.04 **^,#^
GSH-PX (U/mgprot)	879.95 ± 31.91	1064.26 ± 44.89 *	899.63 ± 44.47	1219.64 ± 31.95 **^,#^

Values are given as the means ± SD, *n* = 3. * *p* < 0.05; ** *p* < 0.01 vs. Control group. ^#^
*p* < 0.05 vs. AA I group.

**Table 2 ijms-23-03674-t002:** IC 50 Values of AA I with or without genipin treating transfected or non-transfected cells.

Parameter	UCP2-DNA Transfection (−)	UCP2-DNA Transfection (+)
AA I alone	102.85 ± 7.39	117.43 ± 8.76 *
AA I with Genipin	70.22 ± 5.77 *	92.89 ± 6.27 ^#^

Values are given as the means ± SD, *n* = 3. * *p* < 0.05 vs. AA I transfection (−) group; ^#^
*p* < 0.05 vs. AA I with genipin transfection (−) group.

## Data Availability

The data presented in this study are available on request from the corresponding author.

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
