# Peer review of "Protective Effects of Mitochondrial Uncoupling Protein 2 against Aristolochic Acid I-Induced Toxicity in HK-2 Cells"

_ijms, 2022, doi:10.3390/ijms23073674_

Round 1
Reviewer 1 Report
The Authors have performed most of the suggested experiments. The manuscript has been largely improved.
Author Response
Point 1: The Authors have performed most of the suggested experiments. The manuscript has been largely improved.
Response 1: Thanks for the reviewers' positive comments on our article. We have made the following changes according to your advice:
We improved the introduction section.Some background content has been added, and references cited have also been added.
In the method part, we have made great changes and introduced some important experimental methods in detail.
We also improved the results section to make it easier for readers to understand.
In addition, we added the conclusin section to explain the experimental results in more detail and then draw a conclusion.
Last but not least,We have improved the English grammar and spelling of the whole paper.
Please pay attention to our revised manuscript for details
Reviewer 2 Report
This paper was planned in order to assess whether UCP2 having antioxidative and antiapoptotic properties, could protect HK-2 cells from toxicity induced by aristolochic acid I (AA I).
Even if the topic investigated is of interest and the reported findings have novel aspects, the manuscript needs to be revised before a possible acceptance in Int. J. Mol. Sci.
Firstly, the paper needs an overall revision and check of the English language by a native speacker or by an language editing service. Many grammar mistakes are present into the text as well as many typos (i.e. missed spaces and italics). Moreover, some sentences result quite hard to read and understand.
In addition, the authors have to check if all references have been cited into the text or reported in the references list.
The Abstract section could be further improved.
I suggest to add a Conclusions section (...as well as to improve this part).
Lastly, check the used acronyms and remember to spell all acronyms at first use.
So, after revision the paper has the potential to be published in this journal.
Author Response
Point 1: Firstly, the paper needs an overall revision and check of the English language by a native speacker or by an language editing service. Many grammar mistakes are present into the text as well as many typos (i.e. missed spaces and italics). Moreover, some sentences result quite hard to read and understand.
Response 1: Thank you for your valuable suggestions. We have carefully checked the whole paper, corrected grammatical and spelling mistakes, and also invited native English speakers to revise our paper. We also rewrote these sentences which may be hard to read and understand.
Point 2: In addition, the authors have to check if all references have been cited into the text or reported in the references list.
Response 2: We checked all the references and confirmed that all the references had been cited into the text by EndNote software.
Point 3:The Abstract section could be further improved.
Response 3: We made some minor changes to the abstract, but we couldn't expand it any further because of the abstract is limited to 200 words. We hope to have your understanding.
Point 4: I suggest to add a Conclusions section (...as well as to improve this part).
Response 4: Thanks for your valuable comments. We have added the conclusion part in the revised version
Point 5: Lastly, check the used acronyms and remember to spell all acronyms at first use.
Response 5: Thank you for your kind remind. We have already checked the used acronyms.
Round 2
Reviewer 2 Report
The revised paper is now acceptable.